# Efficient Oil Removal of Polymer Flooding Produced Sewerage Using Super-Hydrophobic Mesh Filtration Method

**Wanli Kang [1,\*], Xin Kang [1], Hongbin Yang [1], Hailu Gebremariam [2] and Zhe Li [2]**

[1] School of Petroleum Engineering, China University of Petroleum (East China), Qingdao 266580, China; xinkang1001@163.com (X.K.); yhb0810@126.com (H.Y.)

[2] Unconventional Petroleum Research Institute, China University of Petroleum (Beijing), Beijing 102249, China; jingxin_2020@126.com (H.G.); lizhecup@126.com (Z.L.)

\* Correspondence: kangwanli@upc.edu.cn; Tel.: +86-135-8933-2193

**Abstract:** During the past 20 years, polymer flooding has become a successful enhanced oil recovery (EOR) technique for mature reservoirs with high water cut and recovery percent around the world. However, the high bulk viscosity of polymer solutions could slow down the separation rate of the crude oil emulsion and make it difficult to treat the produced fluid. Consequently, the efficient removal of oil from the polymer flooding produced sewerage has still drawn significant concern. In this research, a high flux super-hydrophobic copper mesh was prepared using two-stage processes to treat the sewerage from polymer flooding. The surface of the super-hydrophobic mesh was characterized using various techniques including scanning electron microscope (SEM), OCA 20-contact angle goniometer, etc. Accordingly, the static contact angle of the super-hydrophobic copper mesh reached up to 165°. Moreover, the performances of the mesh were systematically evaluated under different internal and external factors such as oil to water volume ratio, polymer concentration, shear rate, and pH. The corresponding configuration and separation mechanisms are further explained in detail. The prepared superhydrophobic mesh can be a potential candidate for sewerage with both a polymer solution and crude oil.

**Keywords:** super-hydrophobic copper mesh; sewerage; polymer flooding; separation mechanism

## 1. Introduction

Since the depletion of petroleum hydrocarbon reservoirs in the world has progressively increased during the past decades, it is believed that enhanced oil recovery (EOR) technologies will play significant roles to secure the growing energy demands in the coming years [1]. Consequently, chemical EOR (cEOR) technologies have received substantial attention from the petroleum industry, among which polymer flooding technology has become one of the most important cEOR technologies [2,3].

During the polymer flooding process, polymer (partially hydrolyzed polyacrylamide, HPAM) is dissolved in water to increase the bulk viscosity prior to water injection into the formation. Hence, the increased water viscosity could lower the mobility ratio of water to oil, thus increasing the volumetric sweep efficiency [4,5]. In addition, some studies have also shown that polymer solutions could increase the displacing efficiency of residual oils [6,7]. Industrial experiences indicate that polymer flooding can increase oil recovery by up to 12% and play a significant role in oil exploitation.

Despite the success of polymer cEOR technologies, the existence of polymer molecules makes it difficult to treat the produced fluid. The sewerage from polymer flooding contains crude oil droplets, suspended solids, organic matters, colloids, paraffins, etc. [8]. The high bulk viscosity of polymer solutions could decrease the separation rate of the crude oil emulsion. Therefore, the efficient removal

of oil from the emulsion is an alarming concern for the environment and economy of the project. Currently, many separation techniques of oil/water or water/oil emulsions have been developed [9] such as gravitation separation [10], floatation separation [11], and chemical coagulation separation [12], etc., whereas they are limited by low efficiency, high energy or chemical consumption, high cost, etc.

To minimize the limitations above-mentioned, the mesh/membrane separation method has been proposed and many types of mesh/membrane materials have been fabricated [13–15]. Nevertheless, membrane materials like polymer membranes cannot be suited for the separation of the crude oil emulsion, especially when it contains polymers. The lifetime of polymer-based membranes could become unacceptably short or their selectivity become unacceptably low as they may be swollen or weaken. Furthermore, the physical structures of most polymer-based membranes may change with the environmental factors like temperature [16,17].

Hence, developing a cost-effective mesh separation material for sewerage generated from polymer flooding produced fluid is still an issue in the petroleum industry. To overcome the limitation of traditional polymer membranes, a super-hydrophobic copper mesh was fabricated using two-stage processes, where nano-needle structures could be grown based on the copper mesh skeleton. The novel metal mesh has good mechanical strength, long using age and high flux, which are more suitable for oil–water separation of polymer-containing produced fluid. The current study focuses on the relationship between the microstructures and the macroscopic separation rules of the super-hydrophobic mesh as well as investigates the basic separation mechanism with the influence of different internal and external factors. The current study will provide theoretical guidance in the ground separation of produced sewerage from polymer flooding.

## 2. Materials and Methods

### 2.1. Materials

Copper meshes with a pore size of 100, 200, and 400 mesh were employed as the substrate and supplied by Dingrun Wire Mesh product Co. Ltd (Beijing, China). It consists of 85–90% copper (mass fraction) and 5–15% tin content. Stearic acid, hydrochloric acid (HCl), anhydrous ethanol, sodium hydroxide (NaOH), and ammonium persulfate $(NH_4)_2S_2O_8$ were all supplied by Sinophram Chemical Reagent Co. Ltd. (Shanghai, China). The crude oil was obtained from Shengli Oilfield (Dongying, China) with apparent viscosity of 50 mPa·s and a density of 0.81 g·cm$^{-3}$ at room temperature (see Table 1). The crude oil consists of dissolved compounds and fine suspended particles. The employed polymer was HPAM, with the molecular weight of $1000 \times 10^4$ g/mol and hydrolysis degree of around 30%. Deionized water with a resistivity of 18.2 MΩ·cm was prepared in the laboratory.

**Table 1.** Properties of the crude oil.

| Properties | Values |
|---|---|
| Viscosity at room temperature (mPa·s) | 50 |
| Density at room temperature (g·cm$^{-3}$) | 0.81 |
| Saturate (%) | 73.1 |
| Aromatics (%) | 14.4 |
| Resin (%) | 6.2 |
| Asphaltene (%) | 7.3 |

### 2.2. Methods

#### 2.2.1. Fabrication of the Super-Hydrophobic Copper Mesh

Figure 1 shows the fabrication of the super-hydrophobic copper mesh. Initially, the etchant solution was prepared by mixing various proportion of $(NH_4)_2S_2O_8$ and NaOH in deionized water. Next, the copper mesh was washed and then dried to remove the native oxide and oil. Then, the mesh was immersed into the mixed solution of NaOH and $(NH_4)_2S_2O_8$ to create a Cu(OH)$_2$ film with rough

micro/nanostructures on the mesh surfaces (room temperature, NaOH concentration = 1.25 mol·L$^{-1}$, (NH$_4$)$_2$S$_2$O$_8$ concentration = 0.63 mol·L$^{-1}$). The mesh was subsequently rewashed using deionized water to displace the etchant solution and dried for 30 min at 70 °C. Finally, the copper mesh was dipped into stearic acid for 60 min to lower the surface free energy. In principle, when the copper mesh is immersed into the etchant solutions, the Cu(OH)$_2$ films micro/nanostructures will be generated on the surface of the mesh. The growth of the Cu(OH)$_2$ films was a function of the reaction time, etchant concentration, and cycle of etching. Stearic acid was used to enervate the surface energy of the mesh. The super-hydrophobicity is attributed to the synergetic effects of both the etchant solution and stearic acid.

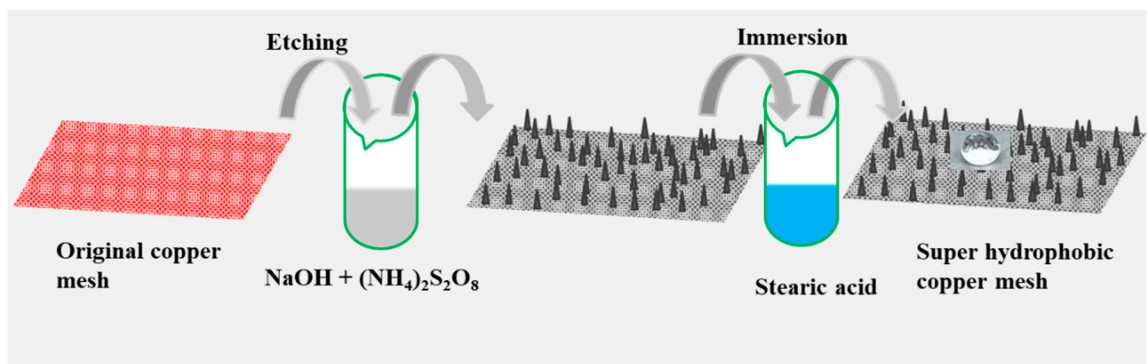

**Figure 1.** Fabrication of the super-hydrophobic copper mesh.

### 2.2.2. Scanning Electron Microscope (SEM)

The micro/nanostructures of the pristine and modified super-hydrophobic copper surface were observed by a Hitachi S-4800 scanning electron microscope (Hitachi Corp., Tokyo, Japan).

### 2.2.3. Contact Angle (CA) Measurement

The static and dynamic contact angle of the super-hydrophobic surface was calibrated at the standard condition with deionized water as a contact liquid using an OCA 20 goniometer (Data Physics, Filderstadt, Germany). Contact angle (CA) measurement was carried out by placing 4-μm deionized water droplets onto the super-hydrophobic surface.

### 2.2.4. Separation Set Up and Separation Efficiency Measurement

As shown in Figure 2, the sewerage enters from the top left half of the separator. Afterward, the sewerage remained inside until most of the oil was removed from the sewerage. The separation efficiency can be calculated as follows:

$$\eta(\%) = \left(1 - \frac{C_P}{C_F}\right) \times 100 \tag{1}$$

where $C_F$ and $C_P$ are the total liquid volume before filtration and the oil volume after filtration, respectively. The permeable oil was collected in a graduated cylinder until every separation finished and the liquid volume was subsequently measured and recorded.

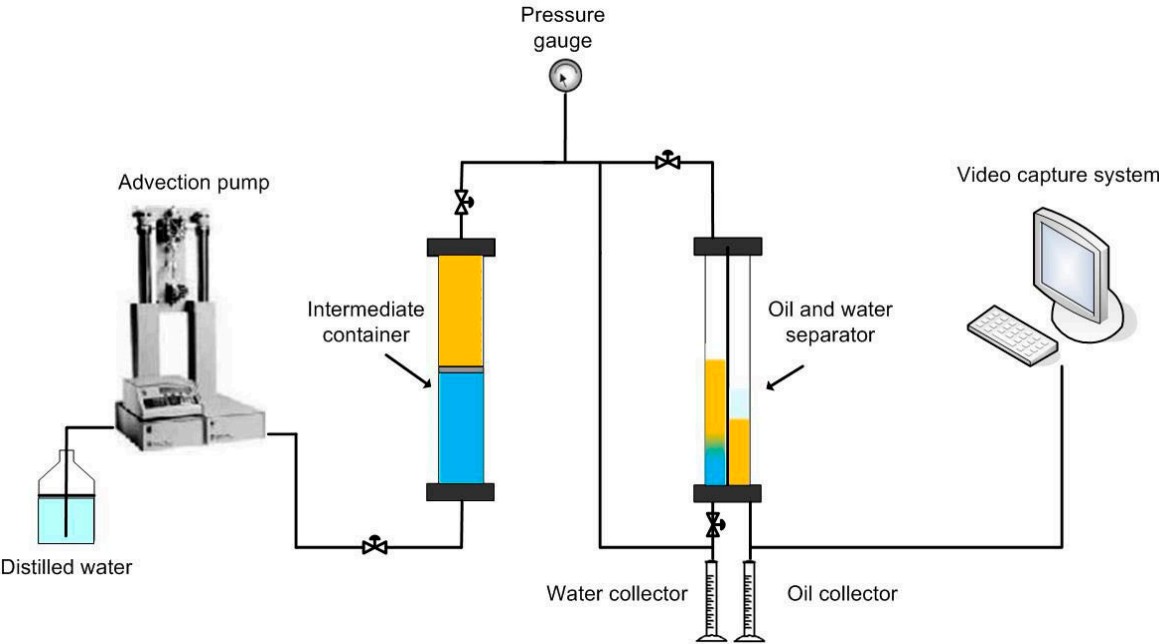

**Figure 2.** Schematic representation of the sewerage treatment process.

## 3. Results and Discussion

### 3.1. Surface Characterization

As shown in Figure 3a, the surfaces of unmodified copper were smooth and no tapering structure was observed in the SEM images, whereas with an increase in the exposure of etching time, the SEM images of the modified copper surfaces showed various densities of micro/nanostructures. An increase in the micro/nano-sized protrusions on the micro-sized particle led to the hierarchical micro/nano binary structures on the $Cu(OH)_2$ film surface. For an immersion time of 60 min, lots of needle-type protrusions with sizes ranging from several hundred nanometers to several micrometers were uniformly formed on the surface of the copper mesh (Figure 3b).

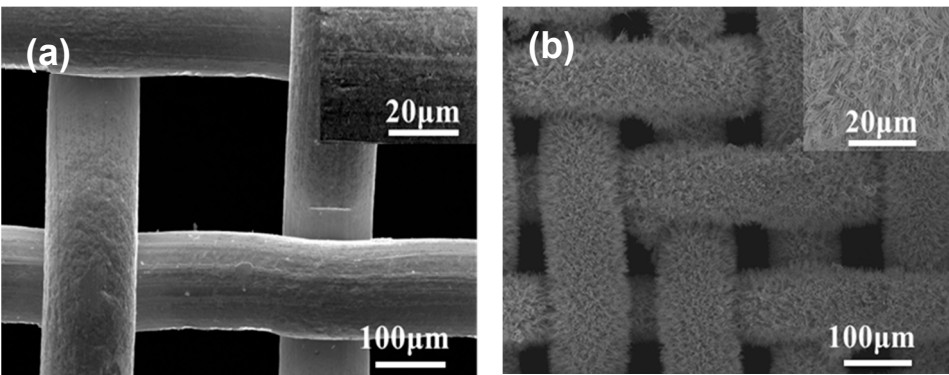

**Figure 3.** Scanning electron microscopy (SEM) images of (**a**) smooth copper mesh and (**b**) 60 min immersion in coating solution.

Wettability of the unmodified and modified copper mesh surface plays a significant role in the separation of oily–wastewater. The water CA of the pristine copper mesh was 63°, which was inherently hydrophilic as a result of the high surface energy of the mesh. After the pristine mesh doused in the etchant solution, the CA of both water and kerosene became zero. When the mesh underwent etching and was then doused in stearic acid, the CA increased noticeably. A maximum static CA of 165° was

recorded for the best mesh with a sliding angle of 2°, suggesting that the bigger CA was acquired with the earlier formation of dense roughness (see Figure 4).

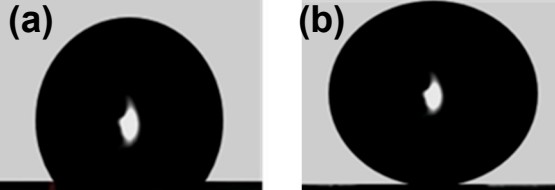

**Figure 4.** Water droplet contact angle on various surfaces after treatment with stearic acid: (**a**) smooth copper mesh surface prior to etching, (**b**) 60-min etching.

### 3.2. Separation Rules of Polymer-Containing Sewerage

The separation processes of sewerage containing polymers and the factors that affect sewerage treatment such as polymer concentration, oil/water volume ratio, the viscosity of crude oil, shear rate, salinity, and pH are investigated and discussed in this section.

Generally, the mixture of water and crude oil is unable to mix at standard conditions because of the hydrophobic effect. Such interaction between oil molecules and water molecules can be mainly driven by the entropy effect. The disruption of highly dynamic hydrogen bonds between the molecules of liquid water by nonpolar molecules resulted in the instability of the mixture. The oil molecules stick together to minimize the surface area exposed to water molecules. However, when a polymer is added in the mixtures of crude oil and aqueous solutions, the formed emulsions are difficult to separate because of the raised solution viscosity. Furthermore, the polymer molecules could be adsorbed onto the oil–water interface, preventing the oil droplets from coalescence. Therefore, the separation of such a mixture could be much more difficult than conventional ones.

The effect of polymer on oil removal efficiency of the super-hydrophobic copper mesh was investigated under various HPAM concentrations. Overall, the crude oil removal efficiency of the mesh considerably decreased with an increase in the polymer concentration (Figure 5). The oil removal efficiencies for the sewerage containing 0–150 ppm HPAM aqueous solutions were all above 97% and most of the oil was removed in the first 10 minutes. In contrast, the oil removal efficiencies of the sewerage with 300 ppm and 1000 ppm HPAM solution were above 90%. A shear rate of rpm was also applied to form kerosene-based sewerage to examine the influence of the polymer on separation efficiency. It was demonstrated in Figure 6 that separation efficiency also decreased with an increase in polymer concentration.

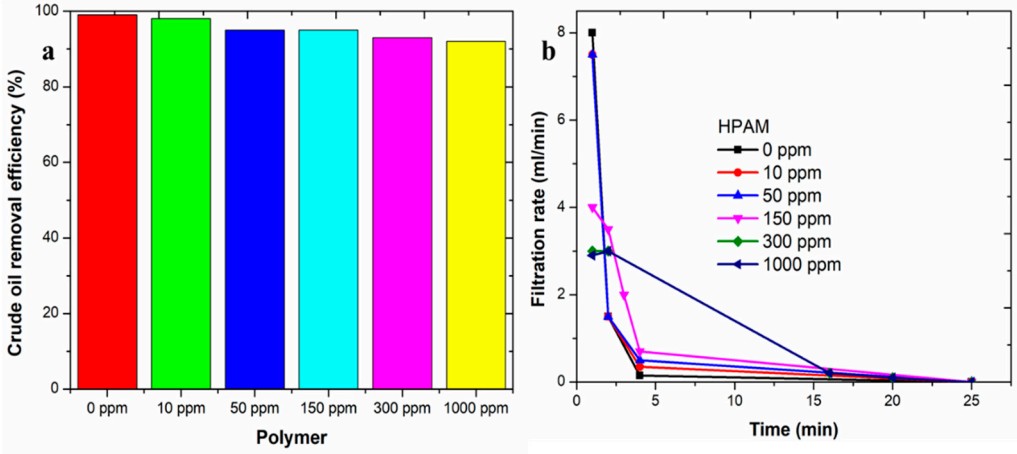

**Figure 5.** Effect of HPAM on the (**a**) oil removal efficiency (**b**) filtration rate, operating condition: O/W = 20/80, HPAM = 50 ppm, shear rate = 5000 rpm.

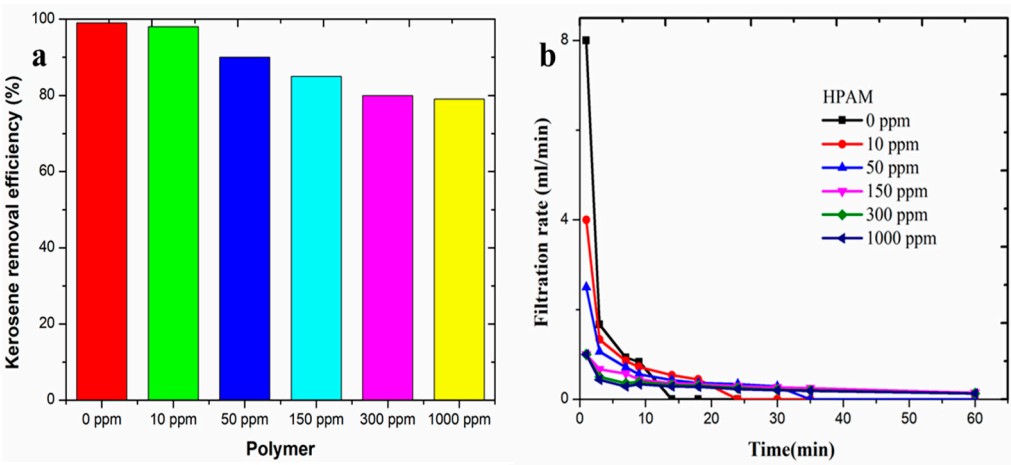

**Figure 6.** Effect of HPAM on the (**a**) oil removal efficiency and (**b**) filtration rate, operating condition: O/W = 20/80, HPAM = 50 ppm, shear rate = rpm.

The zeta potential of the wastewater increased with an increasing concentration of HPAM. It enhanced the total electrical charge density and elasticity of the interfacial film between droplets, so the coalescence of the oil droplet become more difficult under static electrical repulsive force. Accordingly, the emulsion stability was enhanced and oil–water separation became less efficient. However, it could be found that the prepared mesh still separated most of the oil phase, even at a high polymer concentration of 1000 ppm, indicating that the current superhydrophobic can be recognized as a good candidate for mesh separation material.

### 3.3. Effect of Oil/Water Volume Ratio

Another substantial factor that affects the separation efficiency is the oil/water volume ratio of the sewerage. Several tests were carried out using 50 ppm of polymer in oily sewerage samples. Samples were mixed with an oil/water volume ratio of 20:80, 40:60, 50:50, and 60:40, respectively. It can be seen in Figure 7 that when the oil volume fraction was higher, a higher oil removal efficiency could be obtained as well as the filtration rate.

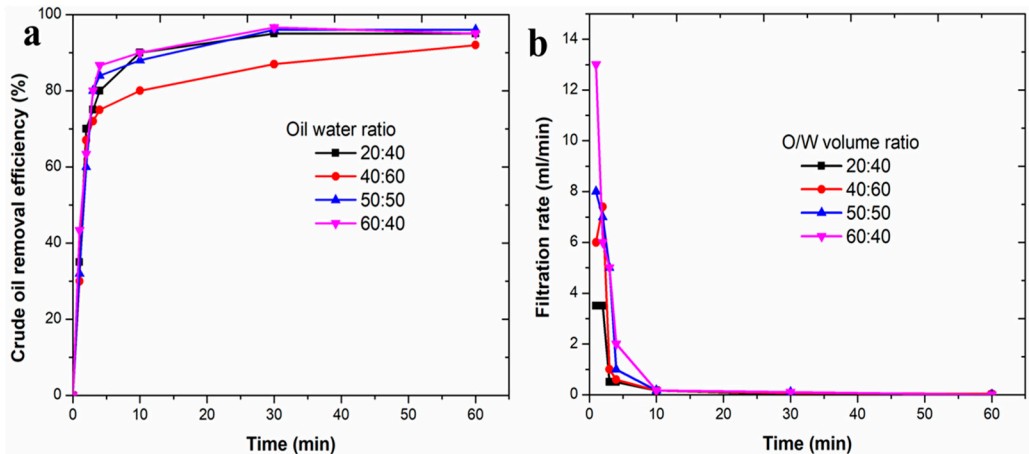

**Figure 7.** Effect crude oil/water volume ratio on the (**a**) oil removal efficiency and (**b**) filtration rate, operating condition: O/W = 20/80, HPAM = 50 ppm, shear rate = 5000 rpm.

The phenomenon could be related to the emulsion type variation. Sewerage can be formed by strongly mixing crude oil and water, which is either a W/O or O/W emulsion. When the continuous phase was water, the contact between the sewerage and separator will be mainly between water and

the super-hydrophobic mesh. Consequently, the lower separation efficiency and separation rate were produced due to the hydrophobic effect.

On the other hand, when the continuous phase was oil, the effective contacting surface area between the mesh and the oil droplets increased, thus it generated a faster oil flow rate. In the mixture of oil and water, the W/O emulsion tended form as a result of a higher oil volume fraction, displaying a higher oil removal efficiency and filtration rate.

### 3.4. Effect of Shear Rate on Sewerage Treatment

During the preparation of wastewater, a sufficient mixing process could ensure that the oil droplets were well dispersed. As shown in Figure 8, the separation rate decreased with increasing shear rate. With a shear rate of 0–4000 rpm, the oil was removed in the first 10 min while it took much more time to remove oil for the 8000 rpm and rpm shear rates. When shear rates were less than 4000 rpm, the removal efficiencies were above 96%. Generally, the higher shear rate was exerted in the formation of sewerage, finer droplets formed, which led to more stable creaming of the wastewater. When the oil droplets were bigger, they could easily coalesce and form oleophilic interaction with the super-hydrophobic copper mesh. Thus, when the oil droplets became smaller, it made coalescence more difficult and there was less interaction with the mesh. In addition, as the shear rate increased, the phase separation was lowered, despite the lowered viscosity (see Table 2), leading to huge interactions between the water and the super-hydrophobic mesh.

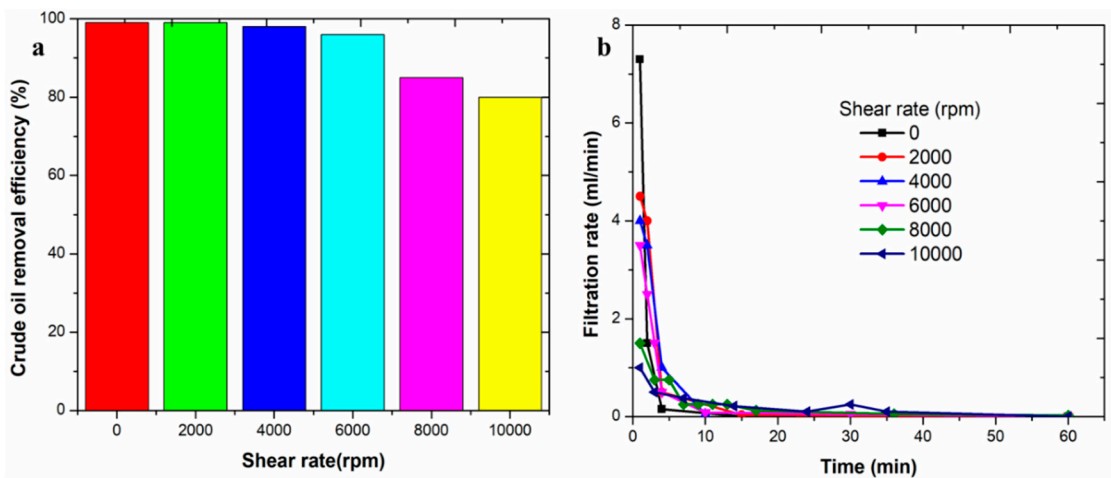

**Figure 8.** Effect of shear rate on the wastewater treatment on the (**a**) oil removal efficiency and (**b**) filtration rate, operating condition: O/W = 20/80, HPAM = 50 ppm.

**Table 2.** Effect of shear rate on the liquid viscosity.

| Shear Rate (rpm) | Liquid Viscosity (mPa·s) |
|---|---|
| 2000 | 42.1 |
| 6000 | 33.6 |
| 10,000 | 13.4 |

### 3.5. Effect of pH on Sewerage Treatment

To investigate the effect of pH on the polymer-based sewerage treatment, acetic acid was added to lower the pH and sodium hydroxide to increase the pH values of the sewerage. Figure 9 shows the effect of pH on oil removal efficiency. Overall, the highest separation efficiencies were recorded at pH values of 4, 5, 11, and 12. However, further lowering the pH led to the dysfunction of the super-hydrophobic mesh. When the pH value increased from 10 to 12, the filtration rate and oil removal efficiency showed a considerable increase. In fact, the pH values affected the wastewater treatment in

two ways. On one hand, pH could affect the properties of the oily–wastewater, which was attributed to the destabilization of oil droplets with pH changes. At extreme pH condition, the functional group in the crude oil of wastewater present in the interfacial film is ionized. Therefore, the mechanical strength is destroyed by introducing internal disturbance in the film. On the other hand, pH had a significant influence on the interfacial properties of crude oil, which was attributed to the fact that pH enhances surface activity as a result of charging the functional group of asphalting.

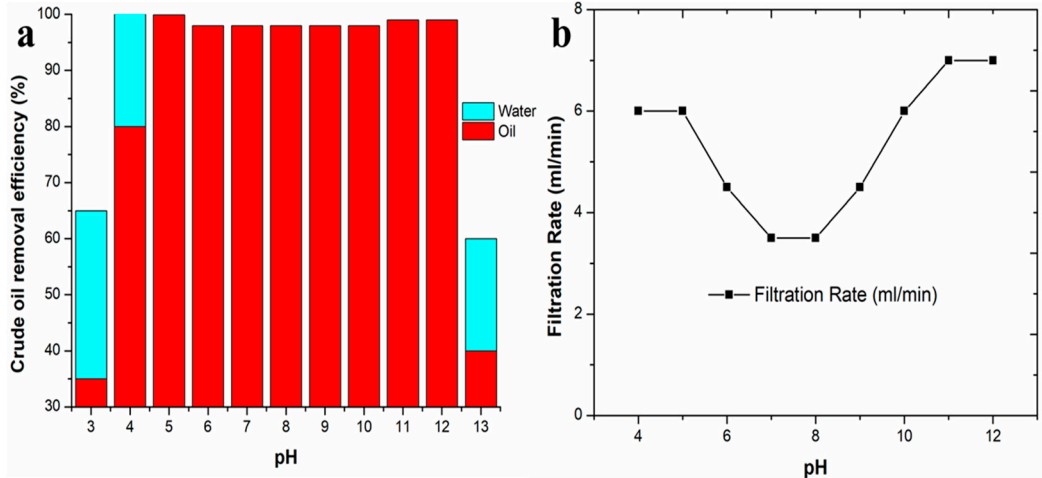

**Figure 9.** Effect of pH on the (**a**) oil removal efficiency and (**b**) filtration rate, operating condition O/W = 20/80, HPAM = 50 ppm, shear rate = 5000 rpm.

In addition, the resin and asphaltene consisting of functional groups that were initially present in the interfacial film were most likely be ionized at extremely low and high pH values. This could change the film properties because of the internal repulsion within the films, which resulted from the higher surface charge density at the interfaces. Furthermore, when acid was added into the oily–wastewater, the viscosity of the oily–wastewater was reduced slightly.

It should also noticed be that both lowering and increasing pH value could cause an ephemeral wettability change under the extreme pH conditions. Since the acetic acid was capable of dissolving in oil and water, it could form a continuous phase of oil or water. When the pH value of the wastewater was lowered to four, the water could penetrate the mesh due to the $H^+$ dissociation on the mesh surface. Further lowering of the pH indicated that the selectively of the mesh had disappeared. Similarly, when the pH values exceeded 12, the filtration efficiency of the mesh was impaired as a result of transient wettability alteration. The strong base NaOH could easily react with stearic acid and weaken the super-hydrophobicity of the mesh. Consequently, the optimal operating pH value for the oily–wastewater treatment ranged from 5 to 12.

### 3.6. Advantage of Super-Hydrophobic Mesh Separation

The stability of the sewerage was investigated under various added HPAM concentrations. Therefore, the gravity-driven role separation and mesh filtration were compared. Figure 10 shows the gravity driven wastewater separation. Overall, the oil–water separation performance decreased with the increase in HPAM concentration.

Figure 11 shows that the phase separation for the sample with no polymer after 1 h was approximately 40%, whereas for samples with a polymer greater than 10 ppm, the phase separation was still less than 5%. After 3 h, the phase separation for samples with <150 ppm (i.e., 0, 10 ppm, and 50 ppm) increased roughly to 90% and the sewerage consisting of 150 ppm and 300 ppm increased to nearly 70%.

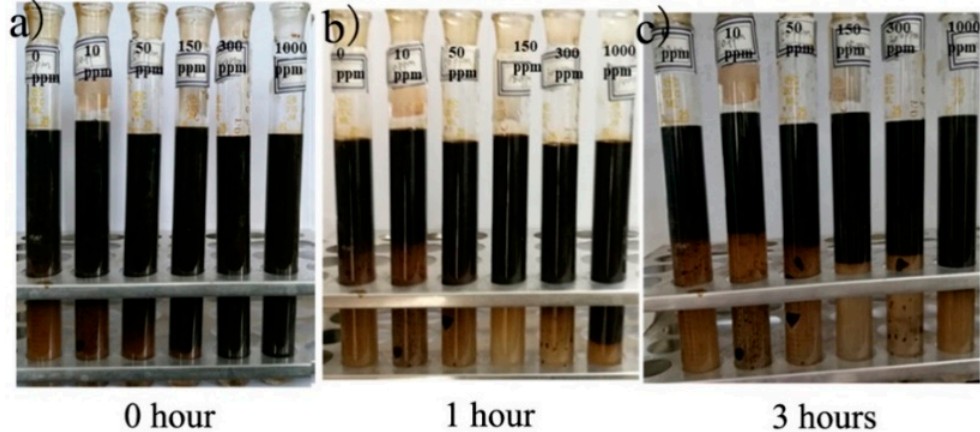

**Figure 10.** Gravity separation process: (**a**) 0.1 h, (**b**) 1 h, and (**c**) 3 h, operating condition O/W = 20/80, HPAM = 50 ppm, shear rate = 5000 rpm.

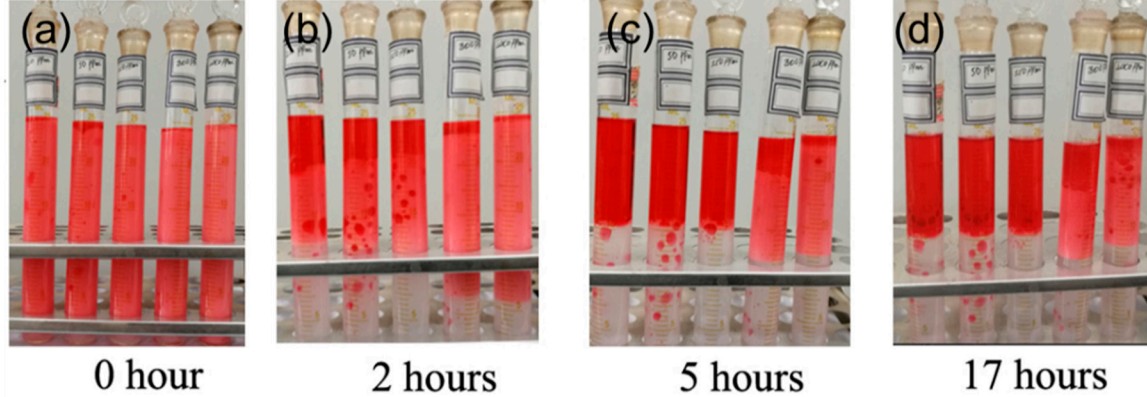

**Figure 11.** Gravity separation process (**a**) 0 h, (**b**) 2 h, (**c**) 5 h, (**d**) 17 h. Operating condition O/W = 20/80, HPAM = 50 ppm, shear rate = rpm.

However, for sewerage containing <150 ppm polymer, most of the oil phase was separated in the first 10 min when using the super-hydrophobic separation technique (see Figure 5). For sewerage with a polymer concentration of 300 ppm and 1000 ppm, it was able to recover more than 90% of the oil fraction from sewerage within 25 min.

Similarly, a gravity separation test was further carried out to evaluate the stability of kerosene-based sewerage. As shown in Figure 11, the phase separation for the samples with 10 ppm, 50 ppm, and 150 ppm were approximately 40%, 30%, and 20% respectively, whereas samples with above 300 ppm, the phase separation were still less than 5% in the first 1 h. After 5 h of delay, the phase separation for samples with less than 150 ppm (i.e., 10 ppm, 50 ppm, 150 ppm) of polymer increased roughly to 90% and the wastewater with of 300 ppm HPAM concentration increased to 20%, while samples with 1000 ppm remained less than 5%. The phase separation for the sample with the highest polymer concentration was recorded as the lowest of all samples, even after 17 h.

In contrast, the mesh separation was faster and nearly 80% of crude oil was recovered within 1 h (Figure 12). Most of the oil phase was separated in the first 20 min for sewerage with a HPAM concentration below 150 ppm. It took nearly 25 min to recover the crude oil from the sewerage containing 300 and 100 ppm polymers.

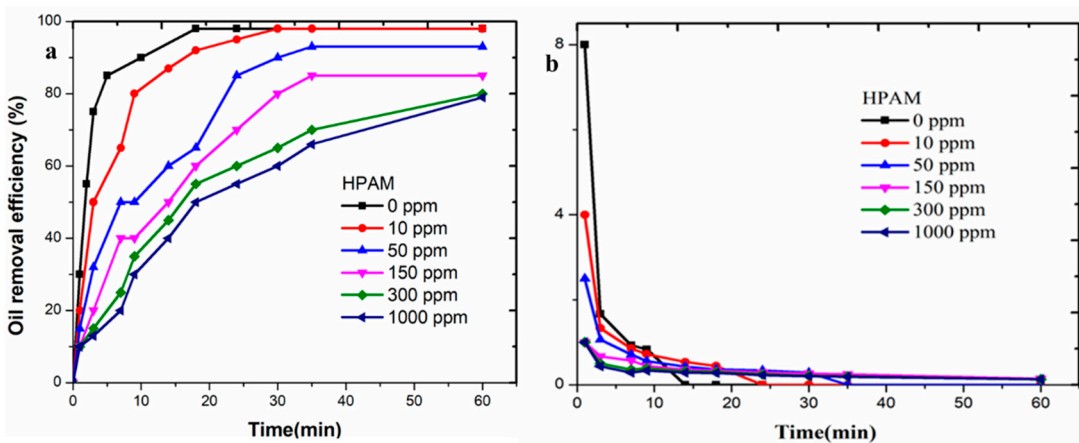

**Figure 12.** Super-hydrophobic mesh separation of the (**a**) oil removal efficiency and (**b**) filtration rate, operating condition O/W = 20/80, HPAM = 50 ppm, shear rate = 5000 rpm.

## 4. Conclusions

In this work, a highly super-hydrophobic and super-oleophilic copper mesh was prepared using two-stage processes to develop a laboratory oil–water separator. Afterward, sewerage separation tests containing polymers were conducted to evaluate the oil removal performances. The oil removal efficiencies for the sewerage containing the HPAM polymer below 150 ppm were above 97% and most of the oil was removed within 10 min. The separation rate and efficiency decreased with increasing shear rate. At extreme pH values, the super-hydrophobic mesh experienced temporary loss of its hydrophobicity and the optimal operating pH values of the super-hydrophobic mesh ranged from 5–12. Furthermore, the current super-hydrophobic copper mesh showed a better separation performance than the traditional gravity separation method, displaying good application potential in the separation of sewerage or emulsions containing polymers.

**Author Contributions:** Supervision, W.K.; Writing—original draft preparation, X.K.; Writing—review and editing, H.Y., H.G., and Z.L. All authors have read and agreed to the published version of the manuscript.

**Funding:** This research was funded by the National Natural Science Foundation of China (No. 51774309) and the National Science and Technology Major Projects of China (2017ZX05009-004).

**Conflicts of Interest:** The authors declare no conflict of interest.

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
