# Peer review of "Efficient Oil Removal of Polymer Flooding Produced Sewerage Using Super-Hydrophobic Mesh Filtration Method"

_colloids, doi:10.3390/colloids4030032_

Round 1

Reviewer 1 Report

The manuscript includes the water/oil separation using the modified mesh. The manuscript is suitable for this journal, but some points should be revised. The comments re as follows,

1) do not use the abbreviation of EOR in abstract.

2) in introduction, based on other researches, please mention the chemical novelty at the last paragraph of introduction. 

3) how to determine the conc. of water and oil.

4) in figure 5, oil efficiency was related to flow rate. please mention the filtration phenomena more concretely using filtration mathematical analysis. did you see the surface of mesh?

5) In view of shear effect, do you have some data of viscosity?

6) Did you have some data of oil species because to explain the pH dependence, asphalting chemicals's information are required.

Reviewer 2 Report

The manuscript under consideration deals with separation problems in EOR field. The paper is well designed providing worth of experimentals trying to show the properties of superhydrophobized copper mesh in separating waste waters ond oil. Few but important remarks require major revison.

-The English language has to be seriously revised containing basic mistakes and words not proper for a scientific and technical work.

-The mesh can be regarded as a strongly heterogeneous surfaces, a mix of large void and irregular solid sirfaces. In these case a static contact angle, if meaningful, should be accompained by hysteresis obtained by advancing-receding contact angle measurement and not only by sliding angle.

-Also roughness measurements should be provided and not qualitatively evaluated

-Beyong wetting properties of pure water, the other liquids included in the study should be tested and compared.

- Polymer details should be reported in Materials section together with a more clear composition of crude oil

-The number of self citations is considerable too high

-Author contribution is not providied

Round 2

Reviewer 1 Report

The manuscript is well revised according to the reviewer's comments, thus at present status is at the level of acceptance.

Author Response

Thanks.

Reviewer 2 Report

The revised version of the paper shows significant improved accordingly with reviewer's comments, nevertheless the manuscript still requires comparative work before publication.

-The English language is still to be carefully checked throughout also in the authors' response

-The polymer details in Materials section are not reported

- A wettability study with HPAM concentrations should help to evidence the coating efficiency

-The SEM pictures included do not help to figure out the roughness conditions
